# Identification of BACE-1 Inhibitors against Alzheimer’s Disease through E-Pharmacophore-Based Virtual Screening and Molecular Dynamics Simulation Studies: An Insilco Approach

**DOI:** 10.3390/life13040952

**Published:** 2023-04-05

**Authors:** Kumarappan Chidambaram

**Affiliations:** Department of Pharmacology and Toxicology, College of Pharmacy, Al-Qara Campus, King Khalid University, Asir Province, Abha 61421, Saudi Arabia; kumarappan@kku.edu.sa

**Keywords:** Alzheimer’s, BACE-1, E-pharmacophore, molecular docking, molecular dynamics simulation, pharmacophore-based virtual screening

## Abstract

Alzheimer is a severe memory and cognitive impairment neurodegenerative disease that is the most common cause of dementia worldwide and characterized by the pathological accumulation of tau protein and amyloid-beta peptides. In this study, we have developed E-pharmacophore modeling to screen the eMolecules database with the help of a reported co-crystal structure bound with Beta-Site Amyloid Precursor Protein Cleaving Enzyme 1 (BACE-1). Flumemetamol, florbetaben, and florbetapir are currently approved drugs for use in the clinical diagnosis of Alzheimer’s disease. Despite the benefits of commercially approved drugs, there is still a need for novel diagnostic agents with enhanced physicochemical and pharmacokinetic properties compared to those currently used in clinical practice and research. In the E-pharmacophore modeling results, it is revealed that two aromatic rings (R19, R20), one donor (D12), and one acceptor (A8) are obtained, and also that similar pharmacophoric features of compounds are identified from pharmacophore-based virtual screening. The identified screened hits were filtered for further analyses using structure-based virtual screening and MM/GBSA. From the analyses, top hits such as ZINC39592220 and en1003sfl.46293 are selected based on their top docking scores (−8.182 and −7.184 Kcal/mol, respectively) and binding free energy (−58.803 and −56.951 Kcal/mol, respectively). Furthermore, a molecular dynamics simulation and MMPBSA study were performed, which revealed admirable stability and good binding free energy throughout the simulation period. Moreover, Qikprop results revealed that the selected, screened hits have good drug-likeness and pharmacokinetic properties. The screened hits ZINC39592220 and en1003sfl.46293 could be used to develop drug molecules against Alzheimer’s disease.

## 1. Introduction

Alzheimer’s disease (AD) is a common type of neurodegenerative disorder that gradually leads to memory loss and cognitive impairment and eventually results in dementia, which is a threatening condition among the elderly population [1], affecting more than 45 million people worldwide and which is expected to increase to 2 trillion people in the next few decades [2,3,4]. The symptoms are progressive cognitive decline, psychosocial behavior disturbances, and memory loss, as well as the presence of senile plaques, neurofibrillary tangles, and decreased cholinergic transmission [5]. In the absence of effective treatments, the number of affected individuals is likely to increase quickly as the population ages. At present, despite recent advances in the treatment of AD, treatment efficacy is small, incurable, and susceptible to drug resistance [6,7]. Based on either acetyl cholinesterase inhibition or cholinesterase inhibitors and N-methyl-D-aspartic acid (NMDA) receptor blockade mechanisms, the existing treatments for promoting the clearance of Aβ or Tau are almost entirely ineffective and also provide only symptomatic relief, underscoring an emergency need for alternative drugs [8]. Plants are repositories for several chemical entities that can provide a significant way to diagnose AD. Although many herbal medicines are undoubtedly effective, only a small number have been clinically used for biological functions, particularly AD. Plant-based phytocompounds provide multifunctional biological applications like anti-neurodegenerative and antioxidant properties [9,10]. In AD, oxidative stress is directly related to the development of pathological mechanisms, namely Aβ accumulation, neurological apoptosis, and tau hyperphosphorylation [11]. Therefore, antioxidants are expected to play an essential role in the efficient treatment of AD and improve cognitive function more successfully [12].

BACE-1, the key enzyme of the amyloidogenic pathway, has an attractive therapeutic target for AD, which involves the production of Aβ and the inhibition of β-secretase [13]. The BACE-1 enzyme plays a vital role in the production of neurotoxic β-amyloid (Aβ) peptides in the brain, which cause AD. The BACE-1 enzyme is highly expressed in the brain and belongs to the family of aspartyl proteases. The literature survey reported that catalytic dyad residues of Asp32 and Asp228 contain BACE-1 proteins involved in maintaining the optimal acidic pH and sequence specificity for enzymatic activity [14]. BACE1 levels and activity rates are elevated in AD brains and bodily fluids, supporting the idea that BACE1 is crucial to the pathogenesis of AD. In order to better understand BACE1 regulation and find additional possible therapeutic targets in the β -secretase pathway, a number of cell biology investigations of BACE1 were conducted. Therefore, the BACE-1 enzyme has attractive drug targets for the development of an anti-Alzheimer’s drug [15].

In this study, we performed E-pharmacophore modeling to identify the potent BACE-1 inhibitor against AD. In this study, four-point pharmacophore features (ARRD) are taken and screened against the eMolecules database. Furthermore, the screened hits are used for structure-based virtual screening. Then the top hits obtained from the structure-based virtual screening [16] are used for MDS and ADME/T prediction to study the stability and pharmacokinetic properties, respectively.

## 2. Materials and Methods

### 2.1. Protein Preparation and Grid Generation

The 3D crystal structure of BACE-1 protein (PDB ID: 6EJ3, 1.94 resolution) was retrieved from the Protein Data Bank (http://www.rcsb.org/ accessed on 12 December 2022). Then, the protein was refined by assigning the proper bond order and charges. After that, hydrogen was added, and the water molecules were removed by the protein preparation wizard (Schrodinger, LLC, New York, NY, USA). The structure was optimized and minimized by the OPLS_2005 force field [17,18]. The grid box was centered at coordinates x = 38.29, y = 59.94, and x = 50.33, respectively.

### 2.2. Molecular Docking and E-Pharmacophore Modeling

The Glide-XP module (Schrodinger, LLC, New York, NY, USA) was employed to predict the binding mode of a bioactive compound in the active site of BACE-1 through molecular docking analysis [19]. The best conformer of the bioactive compounds along with the binding pose was given as input for E-pharmacophore modeling. The E-pharmacophore modeling was constructed based on the energetic terms from the Glide-XP scoring function. The constructed E-pharmacophore was used as input for virtual screening. Pharmacophore sites were automatically created with the phase module (Schrödinger LLC, New York, NY, USA) using six default common chemical features, namely hydrogen-bond acceptor (A), hydrogen-bond donor (D), hydrophobe (H), negatively ionizable (N), positively ionizable (P), and aromatic ring (R) [20,21].

### 2.3. E-Pharmacophore-Based Virtual Screening

Explicit matching is a significant requirement for the most energy-efficient sites in the E-pharmacophore approach, with a score increased from −1.0 kcal mol^−1^. Virtually chosen molecules must match at least 4 hypothesis sites out of 5 or more pharmacophoric sites. The four-point pharmacophore hypothesis (ARRD) was used to screen molecules in the eMolecules database, such as zinc, enamine, life chemical, asinex, and toslab. The screened hits were selected based on their fitness scores, which provide the measurement of hypothesis matching along with aligned ligand conformers depending on the alignments, volume, and RMSD of the hits. Then, the fitness scoring function (0–3) was given during the screening. Screened hits with fitness scores above the value of 0.5 were further filtered by structure-based virtual screening [22].

### 2.4. Structure-Based Virtual Screening

Structure-based virtual screening was performed to find out the novel hits obtained from E-pharmacophore-based virtual screening. The structure-based virtual screening was conducted through the high-throughput virtual screening (HTVS) docking protocol (Glide modules, Schrodinger, LLC, New York, NY, USA). The screening was initially performed by using the HTVS Glide module, and the top-scoring compounds were subjected to standard precision (SP) docking. Finally, the extra precision (XP) docking method was employed to identify the best hits [23].

### 2.5. Enrichment Calculations

The efficacy of virtual screening was assessed using enrichment matrices. Enrichment calculations were performed using the ligands with the best docking score and good binding energy using the leads and decoys from the extra-precision docking file.

### 2.6. Binding Free Energy

The Prime MM-GBSA (Molecular Mechanics-Generalized Born Surface Area) method was used to compute the binding free energy of the protein ligand complexes by using the Prime module (Schrodinger, LLC, New York, NY, USA). The total free energy (ΔG bind) was calculated by the following equation [24]:ΔG_bind_ = ΔG_complex_ − (ΔG_protein_ + ΔG_ligand_)

### 2.7. Molecular Dynamic Simulation (MDS)

A molecular dynamic simulation was performed on protein-ligand complexes by using the GROMACS package with a GROMOS96 43a1 force field. The PRODRG server was used to generate the topology and parameters for the ligands. The complexes were equilibrated under cubic periodic boundary conditions at 1.5 nm, and an explicit SPC (simple point charge) water model was used to solvate the system. The appropriate counterions, 9 Na^+^, were added to neutralize the system. To remove the weak van der Waals contacts of the system, energy minimization was performed for 50,000 steps in 200 ps by using the steepest descent gradient algorithm. The NPT and NVT ensembles were employed for 50,000 steps in 200 and 100 ps, respectively. The temperature and pressure of protein-ligand complexes were stabilized at a temperature of 300 K using Berendsen temperature coupling (Thermostat) and a Langevin barostat, respectively. The equilibration of the systems was performed for 1 ns with a 2 fs integration time step, and all the solute atoms were restrained by a harmonic potential with a force constant of 10 kcal/mol Å. After it had equilibrated, the constant temperature was maintained at 300 K with a coupling time of 0.1 ps. Then, 1 atm of pressure was maintained in 100 ps by using the Parrinello-Rahman algorithm. After that, the MDS was carried out for 50 ns. The trajectories were analyzed to calculate the root mean square deviation (RMSD), root mean square fluctuation (RMSF), and hydrogen bond by using GROMACS utilities [25]. The last 10 ns of the MDS trajectory were assessed using the g_mmpbsa tool (Molecular Mechanics Poisson-Boltzmann Surface Area) for calculating the binding free energy on selected screened hits. The potential energy of molecular mechanics, such as solvent-free energy, was carried out on each complex according to the following equation:ΔG_binding_ = G_complex_ − (G_protein_ + G_ligand_)(1)

### 2.8. Density Functional Theory (DFT)

The DFT analysis was used to determine the molecular electronic features, such as electron density, HOMO density, and LUMO density, to calculate the chemical reactivity of screened hits. In order to understand the inhibitory reaction against BACE-1 protein, the DFT analysis was carried out by Jaguar modules using hybrid DFT with Becke’s three-parameter exchange potential and Lee-Yang-Parr correlation functional (B3LYP) theory with a basis set at the 6–31G* level (Schrodinger, LLC, New York, NY, USA). In this analysis, the Poisson−Boltzmann solver was applied to enumerate the energy in the physiological environment. The highest occupied molecular orbital (HOMO), lowest unoccupied molecular orbital (LUMO), and HOMO-LUMO energy gap were calculated for the screened hits [26].

### 2.9. ADME Prediction

The best identified compounds were used to analyze the absorption, distribution, metabolism, and excretion (ADME) properties by using QikProp. In this procedure, physically significant descriptors and pharmaceutically relevant properties such as QPlogPo/w, QPlogS, QPlogHERG, QPlogBB level, percent human oral absorption, and Lipinski’s rule of five for compounds were predicted [27].

## 3. Results and Discussion

### 3.1. Molecular Docking and E-Pharmacophore

To check the binding mode of the co-crystal ligand conformation, a re-docking protocol was carried out for the target protein prior to the E-pharmacophore modeling. The protein crystal structure of BACE-1 was retrieved and prepared. The co-crystal ligand was re-docked into the active site using the same protocol, resulting in the ligand binding in the same orientation and location. The binding conformation and co-crystal ligand are superimposed (Figure 1A), and the RMSD value is 1.87 Å. According to studies of the crystal structure of the BACE-1-B7T complex, the binding of the ligand was stabilized by three hydrogen bonds, two salt bridges, and pi-pi stacking interactions with the active site residues (Figure 1B).

The E-pharmacophore hypothesis model is an important computational tool to design novel drug-like molecules from the large chemical libraries within a short time period and with limited computational capacities. The goal of the energy-optimized pharmacophore hypothesis is to combine the stereo-electronic properties of the ligand with the energetics of its interactions with the protein structure. The docked complex is given as input for e-pharmacophore modeling and generates the e-pharmacophore hypothesis. Based on Glide XP energy, the four (two aromatic rings (R19, R20), one hydrogen bond accepter (A8), and one hydrogen bond donor (D12)) are obtained, which is depicted in Figure 2. Based on the hypothesis and its score (R14 = −2.20, R20 = −1.76, A8 = −1.90, and D12= −1.30), we used the generated pharmacophore hypothesis for the further analysis.

### 3.2. E-Pharmacophore-Based Virtual Screening

The obtained hypothesis is further used to create a search query to screen against the eMolecules database to identify drug-like molecules with similar pharmacophore features. Table 1 shows the fitness scores for the screened hits through E-pharmacophore-based virtual screening. During the analysis, more than 1000 screened hits are obtained and listed in Table 1. Among them, a total of seven screened hits are selected based on a fitness score above 1.0. The selected potent hits can be used for screening-based virtual screening [28].

### 3.3. Enrichment Calculations

The internal library of 1000 compounds is generated using screened hits from the eMolecules database and 1000 decoys with a molecular weight of 400 Da from the DUD data set. These 1000 screened hits are docked with the active site of the BACE1 protein. To assess the virtual screening technique with the enrichment factor, BEDROC was calculated. The values of parameters such as EF, ROC, and BEDROC are 38.0 (EF 1%), 0.89, and 0.897 (α = 20), respectively, showing that the generated pharmacophore was active and more effective for the screening of small molecule databases.

### 3.4. Structure-Based Virtual Screening

The structure-based virtual screening was performed to identify the potent inhibitor against BACE-1. Based on phase fitness score, the top 1200 screened hits from chemical databases were selected for structure-based virtual screening in HTVS mode. HTVS docking is frequently employed for screening a number of ligands. Compared to SP docking, conformational sampling for HTVS is substantially more constrained and cannot be employed with score-in-place. A total of 150 compounds were eliminated based on the HTVS docking score. However, 150 compounds were still a lot to put in for extra precision mode (XP), so standard precision (SP) mode docking was used to further narrow the options to a maximum of 40 compounds. The SP docking takes less CPU time and has fewer words in its scoring system than the XP docking, which employs explicit-water technology and descriptions. In any grid-based docking method, the receptor is essentially rigid. However, some flexibility can be given by scaling specific regions of the potential, which occurs in SP mode. In essence, the XP technique of docking is used to screen out false positives and improve the correlation between strong poses and high scores. The sampling method of XP mode is based on an anchor and a more precise growth strategy, and it is less forgiving than SP mode. In XP mode, structures are penalized if statistical findings indicate that one or more groups have inadequate solvation. Finally, based on their docking score, seven screened hits were selected from the XP method of docking (Table 2). Among them, ZINC39592220 (−8.182 kcal/mol) and en1003sfl.46293xc (−7.184 kcal/mol) have the highest docking scores when compared with other screened hits. Figure 3 exhibits the binding site interaction between protein-ligand complexes. The ZINC39592220-BACE-1 complex has shown three hydrogen bond interactions with amino acid residues of ASP32 (bond distance 1.87 Å), TYR198 (bond distance 1.97 Å), and ASP228 (bond distance 2.05 Å), one π-π stacking interaction with amino acid residues of ARG128 (bond distance 5.11 Å), and two salt bridge interactions with amino acid residues of ASP32 (bond distance 2.80 Å) and ASP228 (bond distance 3.40 Å). In the en1003sfl.46293xc-BACE-1 complex, four interactions were observed: three hydrogen bond interactions with amino acid residues of TRP76 with a bond distance of 1.98 Å, ILE126 with a bond distance of 1.86 Å, and SER36 with a bond distance of 1.95 Å, and one π-cation interaction with the residue of PHE108 (bond distance = 4.48). Previous literature has reported that the catalytic dyad amino acid residues, such as ASP32 and ASP228, are involved in the formation of hydrogen bonds and hydrophobic interactions that are essential to inhibit BACE-1 activity [29,30]. In the present study, similar catalytic dyad residues (ASP32 and ASP228) are observed that are actively interacting with ZINC39592220 and en1003sfl.46293xc compounds and forming hydrogen bonds and hydrophobic interactions. From this study, it is suggested that both compounds are effectively bound to the active site of BACE-1 and could be potent inhibitors of BACE-1.

### 3.5. Binding Free Energy

Generally, binding free energies are an important marker to enumerate the binding potency of inhibitors to proteins. Screened hits that showed a good docking score were further subjected to a binding energy calculation using the MM-GBSA method. The ZINC39592220-BACE-1 and en1003sfl.46293xc-BACE-1 complexes were selected from the structure-based virtual screening. The results revealed that the ZINC39592220-BACE-1 complex has exhibited binding free energies of −58.83 kcal/mol. The binding free energy for the en1003sfl.46293xc-BACE-1 complex is found to be −56.951 kcal/mol (Table 3). The outcome of binding free energies revealed that the screened hits, such as ZINC39592220 and en1003sfl.46293xc, may firmly bind in the catalytic region of BACE-1 to inhibit the activity of AD.

### 3.6. Molecular Dynamic Simulation (MDS)

Since structure-based virtual screening and binding free energy calculations were conducted, we have analyzed protein and lead compound interactions in the dynamic behavior using molecular dynamic simulation to investigate the structural stability of bound conformations after binding small compounds within the active site of the protein. The two screened hits (ZINC39592220 = −8.182 kcal/mol and en1003sfl.46293xc = −7.184 kcal/mol), with minimum docking scores and the best binding orientations in the protein binding site, were selected for the molecular dynamics simulation analysis for 50 ns. After 50 ns of MDS, the stability of both complexes was evaluated by determining the root mean square deviation (RMSD). The fluctuation of specific residues around their mean position was determined using RMSF to check the dynamic behavior of the amino acid residues during the simulation period. The binding strength of the protein-ligand complexes was analyzed through the H-bond interaction. Figure 4A–C shows the RMSD, RMSF, and hydrogen bonding results for both protein-ligand complexes. Before starting MDS, the RMSD profile of the protein backbone with screened hit complexes was monitored throughout the 50-nanosecond simulation period to analyze the thermodynamic conformational stability. Figure 4A revealed that the RMSD of the BACE-1 backbone has consistent stability throughout the entire simulation period by maintaining a peak of 1.5 nm (indicated in black). During the simulation, the ZINC39592220-BACE-1 complex initially deviated (from 0 to 15 ns), then it stabilized with a RMSD value of 1.52 nm (indicated in red). The en1003sfl.46293xc-BACE-1 complex showed an initial disturbance in stability (from 0 to 28 nm), then it achieved consistent stability after the simulation period (indicated in blue). From the analysis, it is clearly inferred that the ZINC39592220-BACE-1 complex has more stability than en1003sfl.46293xc-BACE-1 due to the form of various interactions observed between the screened hit of ZINC39592220 and the BACE-1 protein. Furthermore, the RMSF was determined to verify the residual fluctuation and the binding efficiency of screened hits with the active site of BACE-1. The RMSF plot is shown in Figure 4B. In the analysis of the RMSF results, it was observed that the maximum and minimum RMSF values are found to be 0.3 and 0.9 nm for ZINC39592220-BACE-1 and en1003sfl.46293xc-BACE-1 complexes, respectively. Moreover, it is apparent that the higher fluctuation observed between the residues from 367 to 404 is due to the presence of a loop or an undefined secondary structure. Therefore, the fluctuation does not affect the stability of either complex.

Hydrogen bonding is a significant feature in maintaining the stability and also determining the bonding strength between the protein-ligand complexes. Using gmx hbond, the interactive hydrogen bond between the BACE-1 protein and the screened hits is computed using simulation trajectories. The predicted results suggest that ZINC22077550 and ZINC32124441 are formed stable hydrogen bond. In this analysis, there are 5 hydrogen bonds observed in the ZINC39592220-BACE-1 complex, but only 3 hydrogen bonds are obtained in the en1003sfl.46293xc-BACE-1 complex. All the H-bonds are consistently stable throughout the 50 ns simulation (Figure 4C1,C2).

### 3.7. MM-PBSA Calculation

An efficient MM-PBSA calculation method was employed to assess the stability of binding screened hits along with the free energy simulation. According to the results, the compounds ZINC39592220 and en1003sfl.46293xc have the lowest binding energies of −186.228 ±17.577 kJ mol^−1^ and −154.541 ± 22.658 kJ mol^−1^, respectively. It supports the stronger binding stability of ZINC39592220 (−153.644 ± 15.406 kJ mol^−1^, −16.325 ± 13.221 kJ mol^−1^, and −11.430 ± 1.400 kJ mol^−1^) and en1003sfl.46293xc (−158.574 ± 27.587 kJ mol^−1^, −28.258 ± 10.574 kJ mol^−1^, and −16.744 ± 12.647 kJ mol^−1^) for the last 10 ns of the MDS trajectory. The total binding free energy is positively influenced by the polar solvation energy. As a result, the predicted binding free energy strongly supports the binding interaction of newly screened compounds in the targeted protein’s active site.

### 3.8. DFT Analysis

To understand the surface electronic properties of BACE-1 inhibitors, it was necessary to calculate the electronic structure of the screened hits obtained from the structure-based virtual screening. This helped to identify the key functional groups and their surfaces, which are crucial for the interactions. The electronic properties of screened hits (ZINC39592220 and en1003sfl.46293xc) were characterized using frontier orbital energies such as HOMO and LUMO. From this analysis, intermolecular charge transfer between the acceptor and donor functional groups of the screened hits was demonstrated. The calculated stereoelectronic properties are given in Table 3, and the HOMO and LUMO contour maps are depicted in Figure 5. In the ZINC39592220 screened hit, HOMO regions are positioned at the surrounding N-(4-bromo-2-fluorophenyl) formamide group (energy value 0.237), while LUMO regions are presented at the 3-(1H-pyrazol-5-yl)-1,5-dihydro-4H-1,2,4-triazol-4-amine group (energy value −0.094 eV). Based on the energy value and distribution of the HOMO and LUMO regions, HOMO has higher stability when compared to LUMO, suggesting that the ZINC39592220 hit may favor a nucleophilic reaction. In the en1003sfl.46293xc screened hit, both HOMO and LUMO regions are occupied in the benzofuran group, and the HOMO regions have higher stability than that of the LUMO regions based on the energy values (−0.217 eV for HOMO and −0.191 eV for LUMO), suggesting that the en1003sfl.46293xc hit may favor nucleophilic reactions. Moreover, the smaller HOMO-LUMO energy gap (−0.143 eV) is obtained for the ZINC39592220 hit rather than for the en1003sfl.46293xc hit, confirming the higher stability and chemical reactivity of the molecules [31]. The oral bioavailability of the screened hits was computed using the parameter called aqueous solubility. The solvation energies of the screened hits were found to be −25.40 kcal/mol (ZINC39592220) and −20.29 kcal/mol (en1003sfl.46293xc) (Table 4). The higher negative value indicates that the higher aqueous solubility could help improve the drug-like properties of the ligands. From these results, the screened hits have good aqueous solubility. The study confirmed that the developed E-pharmacophore hypothesis was suitable for determining screened hits that could be correlated to the desired inhibitors of BACE-1 activity.

### 3.9. ADME Properties

The screened hits were further screened through the ADME properties to check for any that could possibly be toxic or might be absorbed by the body, rendering it inactive before passage through the membrane. Most of the drug failures were due to the undesirable pharmacokinetic properties and toxicity profiles of the compounds. The effectiveness of therapeutically screened hits depends mainly on their bioactivity and pharmacological properties. The evaluation of the pharmacokinetic and drug-like properties of screened hits is necessary to find a potent safe side and an effective therapeutic potential. The ADME properties of ZINC39592220 and en1003sfl.46293xc compounds were computed using the Qikprop module, and the results are shown in Table 5. The table revealed that two screened hits (ZINC39592220 and en1003sfl.46293xc) obeyed the Lipinski rule of five with a molecular weight of 288 and 255.35 (>500 KD), respectively, a hydrogen bond acceptor of 5 and 9 (>10), respectively, and a hydrogen bond donor of 3.00 and 5.00 (>5.00), respectively [32]. The screened hits showed good partition coefficient (QPlogPo/w) values of −1.204 and −2.001, respectively, which are important for the acceptance of absorption and distribution of drug molecules. The QPPCaco values representing the permeability of the screened hits are found to be −4.877 and −3.305, respectively, which is a key feature for estimating cell permeability in biological membranes. In this study, the selected ZINC39592220 and en1003sfl.46293xc compounds are within the acceptable range of pharmaceutically relevant descriptors, pharmacokinetic profiles, and drug-like properties. Based on the results obtained from pharmacokinetic parameters, it is confirmed that the screened hits have good bioavailability and pharmacological properties and did not violate Lipinski’s RO5 [32]. Therefore, the identified screened hits can be considered potent inhibitors for AD.

In the present study, the results from the E-pharmacophore hypothesis, pharmacophore-based virtual screening, structure-based virtual screening, binding free energy, molecular dynamic simulation, DFT calculation, and ADME properties showed similar bonding patterns with better binding affinity in a stable binding orientation and good drug-like properties. Therefore, the screened hits will help in the further development of a potent and effective drug for the treatment of Alzheimer’s disease through the inhibition of BACE-1.

## 4. Conclusions

According to several studies, Aβ is the main pathophysiologic factor in Alzheimer’s disease. BACE-1 is the main enzyme responsible for amyloidogenic cleavage of APP, which leads to Aβ deposition and plaque formation in the human brain. Therefore, reducing Aβ through control of BACE-1 activity is a sensible therapeutic goal to treat Alzheimer’s disease. The availability of the crystal structures bound with the inhibitor of BACE-1 was used to generate the E-pharmacophore models based on binding mode and interaction energy to concede the inhibitor. Hence, in this investigation, an E-pharmacophore model was employed to analyze the crystal structures of BACE-1 with various filters that included E-pharmacophore-based virtual screening, structure-based virtual screening, binding free energy, molecular dynamics simulation, and ADME properties to identify new BACE-1 inhibitors against Alzheimer’s disease. Systematic comparisons revealed that E-pharmacophore modeling has advantages in finding potent hits with the structural requirements of BACE-1. The results demonstrated that E-pharmacophore models might accurately represent the binding modes of the active site of BACE-1 used in the E-pharmacophore-based virtual screening and structure-based virtual screening. The binding mode analyses of the protein-ligand complexes also revealed the key amino acids in BACE-1 that are required for ligand binding. Based on the above results, it can be proposed that the screened hits ZINC39592220 and en1003sfl.46293xc compounds have more potential to bind with the catalytic dyad region of the BACE1 protein. Also, the MDS and binding free energy study complexes revealed strong dynamics behavior with stable and greater binding affinity. Also, the DFT and ADME properties analysis expresses the effect of frontier orbital energies, including the electrostatic potential of the newly screened inhibitors with inhibitory activity that have good pharmacological properties and did not violate Lipinski’s role. Therefore, it can be concluded that the screened hits might be effective and safer lead molecules for the treatment of AD.

## Figures and Tables

**Figure 1 life-13-00952-f001:**
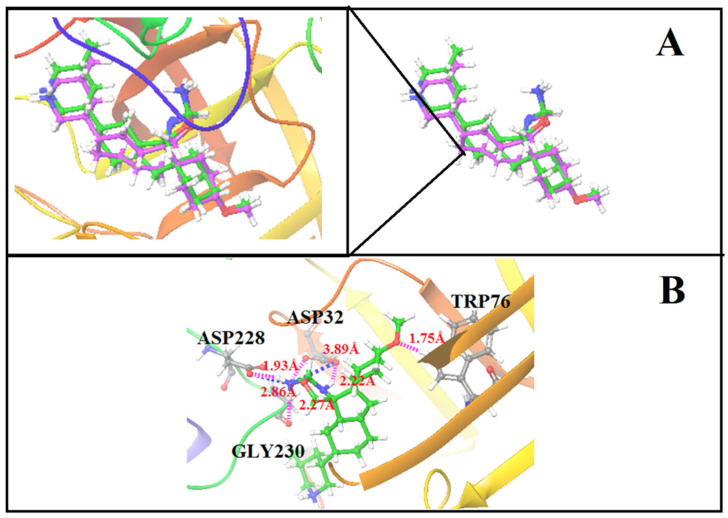
(**A**) Superimposed structure of binding conformation and co-crystal ligand. (**B**) Docked conformation of the co-crystal ligand into the active site of BACE-1.

**Figure 2 life-13-00952-f002:**
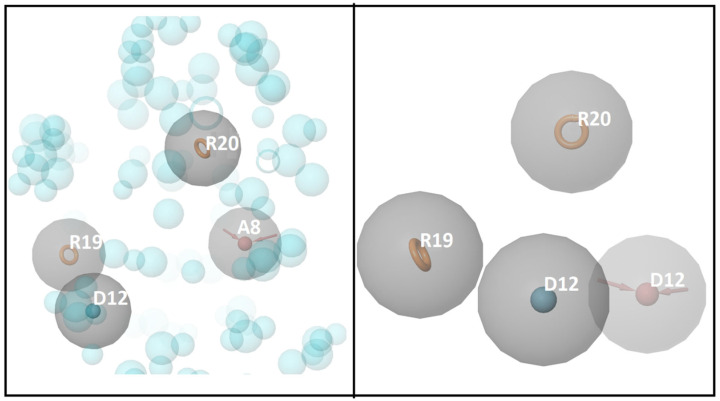
E-pharmacophore hypotheses ARRD.

**Figure 3 life-13-00952-f003:**
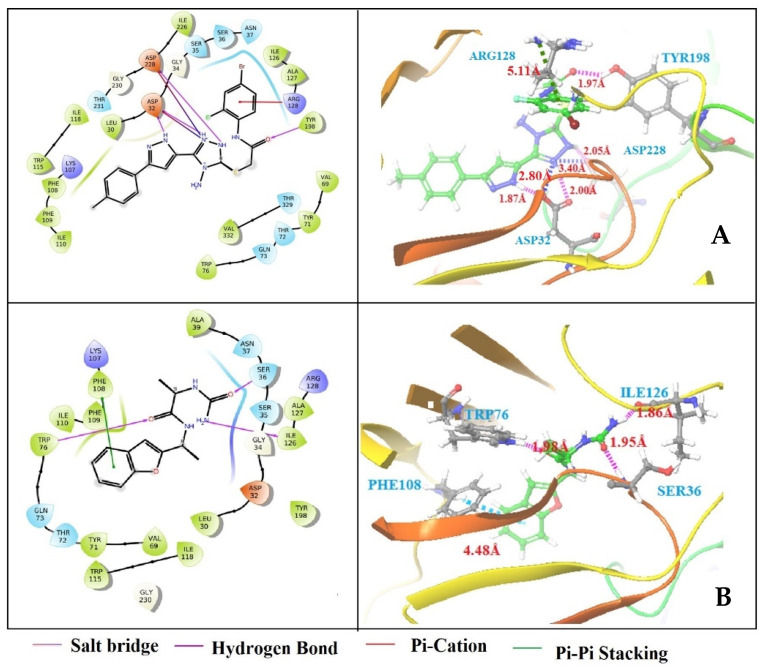
Binding site interaction between BACE-1 and E-pharmacophore-based screened hits from the eMolecules database: (**A**) ZINC39592220 and (**B**) en1003sfl.46293xc.

**Figure 4 life-13-00952-f004:**
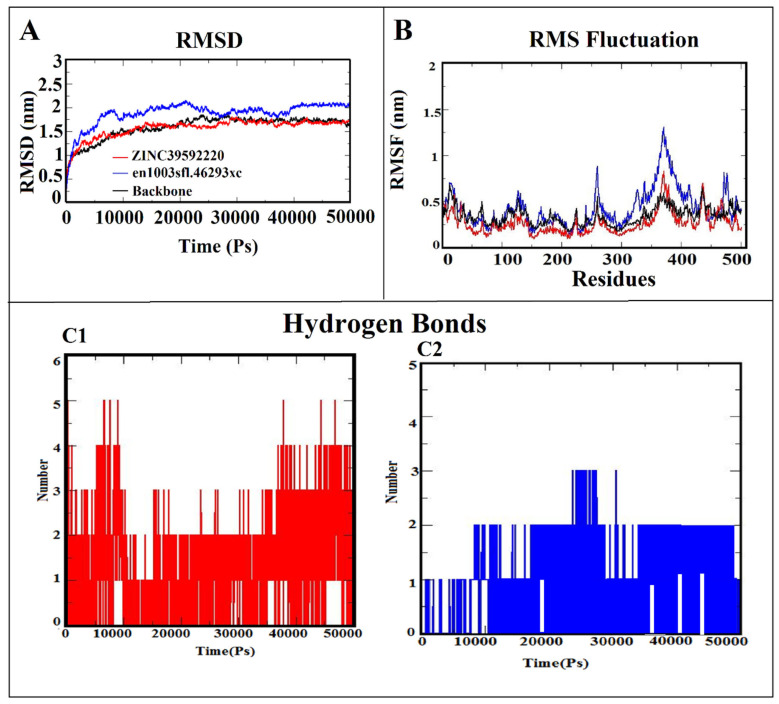
(**A**) RMSD, (**B**) RMSF, and hydrogen bond interaction of (**C1**) ZINC39592220-BACE-1 and (**C2**) en1003sfl.46293xc-BACE-1 complexes.

**Figure 5 life-13-00952-f005:**
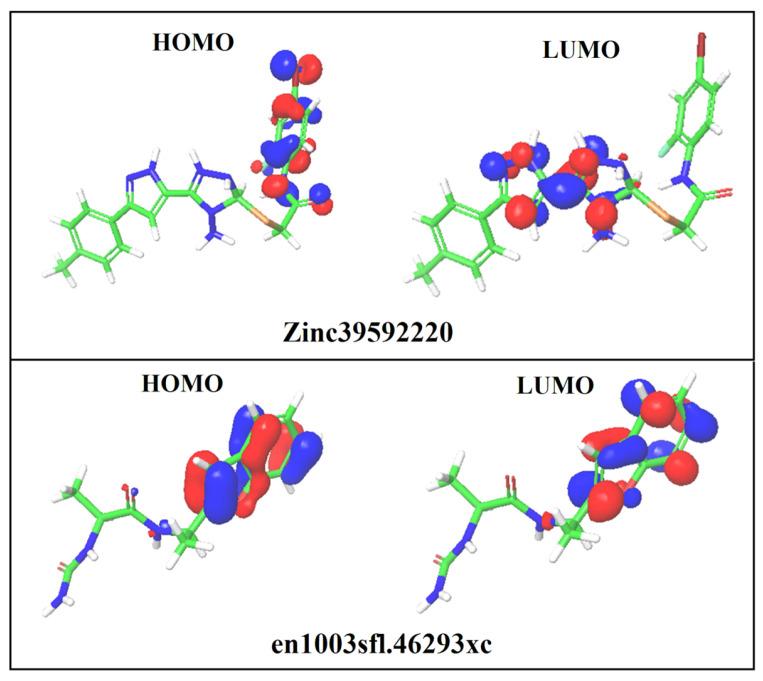
The HOMO and LUMO contour maps of the screened hits.

**Table 1 life-13-00952-t001:** Fitness scores for the screened hits through the E-pharmacophore hypothesis.

S. No	Compound	Site Score	Vector Score	Volume Score	Fitness Score
1	ZINC39592220	0.824	0.837	0.474	1.624
2	LEG 18146402	0.567	0.832	0.359	1.575
3	SYN 19994146	0.434	0.847	0.277	1.571
4	F0513-0458	0.924	0.957	0.345	1.481
5	LEG 18141414	0.840	0.815	0.387	1.432
6	10236	0.499	0.740	0.254	1.413
7	en1003sfl.46293xc	0.818	0.855	0.299	1.399

**Table 2 life-13-00952-t002:** Docking score and glide energy (kcal/mol) of screened hits with BACE-1.

S. No	Compound ID	XP Docking Score (kcal/mol)	Glide Energy (kcal/mol)	∆G Bind kcal/mol
1	ZINC39592220	−8.182	−59.425	−58.803
2	en1003sfl.46293xc	−7.184	−55.284	−56. 951
3	LEG 18146402	−6.933	−49.358	−45.351
4	SYN 19994146	−6.581	−50.369	−48.259
5	F0513-0458	−6.358	−48.259	−50.258
6	LEG 18141414	−5.982	−42.357	−50.592

**Table 3 life-13-00952-t003:** The predicted binding free energy estimates (kcal/mol) of screened hits with BACE-1.

S. No	Databases	Compound ID	∆G Bind kcal/mol
1	ZINC	ZINC39592220	−58.803
2	Enamine	en1003sfl.46293xc	−56.951

**Table 4 life-13-00952-t004:** Molecular electronic features of the screened hits.

S. No	Compound ID	HOMO (eV)	LUMO (eV)	EHOMO–ELUMO(eV)	Solvation Energy (kcal/mol)
1	ZINC39592220	−0.237	−0.094	0.143	−25.40
2	en1003sfl.46293xc	−0.217	−0.026	−0.191	−20.29

**Table 5 life-13-00952-t005:** The ADME properties of the screened hits.

S. No	Compounds ID	MW	HBD	HBA	%HOA	Qplog Po/w	QPP Caco
1	ZINC39592220	288	3	5	93.01	−1.204	−4.877
2	en1003sfl.46293xc	255.35	5	9	75.360	−2.001	−3.305

MW—Molecular Weight of the Molecule (130.0 to 725.0); HBD—Hydrogen Bond Donor (0.0 to 6.0); HBA—Hydrogen Bond Acceptor (2.0 to 20.0); QPlogP o/w—Predicted Octanol/Water Partition Coefficient (−2.0 to 6.5); QPPCaco—Predicted Caco Cell Permeability (<25 is poor, >500 is great); % HOA—Percentage of human oral absorption (>80% is high, <25% is low).

## Data Availability

No data was used for the research described in the article.

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
