# Peer review of "Identification of BACE-1 Inhibitors against Alzheimer’s Disease through E-Pharmacophore-Based Virtual Screening and Molecular Dynamics Simulation Studies: An Insilco Approach"

_life, 2023, doi:10.3390/life13040952_

Round 1

Reviewer 1 Report

I found this work interesting but have some concerns regarding the studies.

1.      The abstract needs to be polished.

2.      I would like to mention that the writing part needs improvement. I feel that the use of phrases could be simplified for easy understanding. Please carefully proof-read spell check to eliminate grammatical errors.

3.      In the abstract, the author has to include results of best molecules scores such as dock energy, free energy and Stable range.

4.      Author give the gird box Size (XYZ) coordinates.

5.      What did you observe from the analysis of the Molecular Dynamics Simulation and how it will be correlated with further studies

Author Response

Dear Reviewer,

Please find the attached response to your comments.

Thanks

Dr Kumar

Reviewer 2 Report

The author has presented a virtual screening protocol to identify the BACE1 inhibitors. This protocol includes screening, docking, and molecular dynamics simulation; however, this work does not include several details and needs a major revision. 

Remarks:

1. On page 4, line 147, the author mentioned that they used Glide XP energy to come up with a four-point pharmacophore hypothesis but did not provide much explanation. A detailed explanation would be absolutely necessary as the screened ligands are based on the four-point pharmacophore hypothesis.

2. The author presented two molecules ZINC39592220 and en1003sfl.46293xc and both of them contain 3 to 5 hydrogen bond donors and 5 to 9 hydrogen bond acceptors which shows these molecules might not have good pharmacokinetic properties. 

3. The author’s virtual screening protocol presented in this paper is not rigorous and not likely to lead to drug-like or lead-like compounds. The author should look into current virtual screening protocols such as Int. J. Mol. Sci. 2020, 21(10), 3626 or J. Chem. Inf. Model. 2022, 62, 1, 116–128. 

Author Response

(The authors gave the same response as above.)

Reviewer 3 Report

In the submitted manuscript Chidambaram developed the E-pharmacophore modeling to screen the eMolecules database against Beta-Site Amyloid Precursor Protein Cleaving Enzyme 1 (BACE-1), an important therapeutic target for Alzheimer's disease (AD). Potent inhibitors of BACE-1 were investigated by applying E-Pharmacophore-based virtual screening, Structure-based Virtual Screening, Binding Free Energy, Molecular Dynamics Simulations, and ADME analysis. Top hits, namely ZINC39592220 and en1003sfl.46293, were selected based on the most negative docking scores, binding free energies, stability throughout MD simulations as well as drug-like pharmacological and pharmacokinetics properties.  

The subject of this manuscript is relevant to the field; however, there are a few comments that should be addressed before the final publication of the manuscript:

Introduction

Lines 42-39: The author nicely reviewed the pathology of AD. However, the literature on plant-based natural compounds with already observed inhibitory effects on AD is not provided. The already known inhibitors of BACE-1 should also be specified.

Section Materials and Methods

2.3. E-pharmacophore Based Virtual screening

The Four-point pharmacophore hypothesis (ARRD) applied to screen the eMolecules database should be described in more detail. The specific parameters, based on which the best hits were selected, should be specified.

2.6. Molecular Dynamic Simulation (MDS)

How many Na+ and Cl- ions were added for the neutralization of the system?

The details of the applied thermalization and equilibration protocol should be provided and properly referenced.

Section Results and discussion

3.1. Molecular docking and E-pharmacophore

Figure 1: I strongly advise the author to present the observed intermolecular interactions in 3D instead of 2D figures. The colors of depicted intermolecular interactions should be explained in Figure caption.

Figure 2: 2D representation of E-pharmacophore hypothesis must be improved. The pharmacophores should be presented on the structural formula of the co-crystal ligand, whose name should also be provided.  

Line 159: The term »Screening Based Virtual Screening« should be revised into »Structure-Based Virtual Screening«.

3.3. Structure-Based Virtual Screening

Subsection 3.3 Structure-Based Virtual Screening should be thoroughly revised.

The author state »In this screening, thirty compounds are selected based on the Docking Score. Here, seven compounds are obtained based on the Docking Score (Table 2).« However, in this subsection, Table 2 with the corresponding results is not provided.

Moreover, the lengths of the observed intermolecular interactions of the two most potential BACE-1 inhibitors should be specified. Which geometrical criteria were used to define intermolecular interactions?

Figure 3: I strongly advise the author to present the observed intermolecular interactions in 3D instead of 2D figures. The colors of depicted intermolecular interactions should be explained in the Figure caption.

3.4. Binding Free Energy

Table 2 should be renamed into Table 3.

3.5. Molecular Dynamic Simulation (MDS)

The conformational stability of ZINC39592220-BACE-1 and en1003sfl.46293xc-BACE-1 complexes was investigated in 50 ns MD simulation. I suggest the author also apply more rigorous binding free energy calculations based on the Linear interaction energy method (LIE), which provides better approximations of the experimental binding free energies. Please inspect the references:

1. Furlan, Veronika, and Urban Bren. (2021) Insight into Inhibitory Mechanism of PDE4D by Dietary Polyphenols Using Molecular Dynamics Simulations and Free Energy Calculations. Biomolecules 11.3, 479.

2. Pantiora, P.; Furlan, V.; Matiadis, D.; Mavroidi, B.; Perperopoulou, F.; Papageorgiou, A.C.; Sagnou, M.; Bren, U.; Pelecanou, M.; Labrou, N.E. Monocarbonyl Curcumin Analogues as Potent Inhibitors against Human Glutathione Transferase P1-1. Antioxidants 2023, 12, 63.

Average RMSD and RMSF values of BACE-1 backbone atoms should be provided and compared. Additionally, the stability of both ligands at the active site of BACE-1 could be assessed through ligand RMSD and RMSF analysis.

Line 217: Please revise ».. of undesired region« into »undefined secondary structure«.

The representation of the graph in Figure 4c) should be modified to eliminate the overlap between red and blue colors. Currently, it is only clear, that there are 2 H-bonds consistently present during MD simulation of ZINC39592220-BACE1 complex.

3.6. DFT analysis

The obtained results of DFT analysis should be correlated with pharmacophore analysis. The identified electron donors should be specified and the rationale for the inhibitory activity clearly presented.

3.7. ADME properties

The acceptable range values for individual pharmacological parameters should be provided.

Author Response

(The authors gave the same response as above.)

Round 2

Reviewer 2 Report

The author has responded to each question well and incorporated those changes in the manuscript. They identified two hits from their screening effort and further analyzed them through molecular dynamics simulation, computation of physicochemical properties, and electronic properties. The whole manuscript has undergone a major overhaul and now it contains sufficient explanations of each method and its results.